# The Problem of Halitosis in Prosthetic Dentistry, and New Approaches to Its Treatment: A Literature Review

**DOI:** 10.3390/jcm10235560

**Published:** 2021-11-26

**Authors:** Magdalena Wyszyńska, Monika Nitsze-Wierzba, Ewa Białożyt-Bujak, Jacek Kasperski, Małgorzata Skucha-Nowak

**Affiliations:** 1Unit of Dental Material Sciences, Department/Institute of Prosthetic Dentistry and Dental Material Sciences, Division of Medical Sciences in Zabrze, Medical University of Silesia in Katowice, 15 Poniatowskiego Str., 40-055 Katowice, Poland; ebialozyt@sum.edu.pl; 2Department/Institute of Prosthetic Dentistry and Dental Material Sciences, Division of Medical Sciences in Zabrze, Medical University of Silesia in Katowice, 15 Poniatowskiego Str., 40-055 Katowice, Poland; mnitsze@sum.edu.pl (M.N.-W.); protstom@sum.edu.pl (J.K.); 3Unit of Dental Propedeutics, Department of Conservative Dentistry with Endodontics, Division of Medical Sciences in Zabrze, Medical University of Silesia in Katowice, 15 Poniatowskiego Str., 40-055 Katowice, Poland; mskucha-nowak@sum.edu.pl

**Keywords:** halitosis, fetor, mouth, malodor

## Abstract

The aim of this work is a review of recent scientific literature about intra-oral halitosis among patients using dentures. Halitosis is a condition in which an unpleasant smell comes out of the oral cavity, which is caused mainly by volatile sulfur and non-sulfured compounds. The etiology of halitosis may be intra- or extra-oral; in most patients, however, it is due to the activity of microorganisms in the oral cavity. The problem of the occurrence of intra-oral halitosis has accompanied patients for many years. In dental prosthetics, the problem of halitosis occurs in patients using removable or fixed dentures. In both cases, new niches for the development of microorganisms may be created, including those related to intra-oral halitosis. It should be noted that dentures—both fixed and removable—are a foreign body placed in the patient’s oral cavity which, in case of insufficient hygiene, may constitute a reservoir of microorganisms, causing this unpleasant condition. Conventional treatment of intraoral halitosis reduces microbial activity via chemical and/or mechanical action. Currently, the search for new strategies in the treatment of halitosis is in progress. One idea is to use photodynamic therapy, while another is to modify poly(methyl methacrylate) (PMMA) with silver and graphene nanoparticles. Additionally, attempts have been made to combine those two methods. Another unconventional method of treating halitosis is the use of probiotics.

## 1. Introduction

Today, in the age of broadly understood social media and the aesthetic standards they create, along with the great need to be socially accepted by meeting them, many people also struggle with exclusion due to halitosis, i.e., bad breath. The prevalence of this ailment in the general population is as high as 31.8% [1]. Halitosis (Latin: fetor ex ore; fetor oris) os a condition in which an unpleasant odor comes out of the mouth, which may be caused volatile compounds of sulfur—such as hydrogen sulfide, methyl mercaptan and dimethyl sulfide—as well as other volatile compounds, such as indole, skatole, acetic acid, propionic acid, and amines (cadaverine and putrescine) [2]. The classification divides halitosis into two groups: genuine halitosis, and delusional halitosis [2,3]. Genuine halitosis, resulting from nighttime hyposalivation, saliva stagnation, and the correction of residual and exfoliated possibilities of food supplementation for bacteria accumulating at the back of the mouth—including pathogenic bacteria—can be of intra- or extra-oral origin [2,4]. Delusional halitosis is a pseudo-halitosis—which is a patient’s subjective feeling that they suffer from halitosis, but it is not clinically tangible—or halitophobia, which is the fear of bad breath [2,3,4]. This classification, by Miyazaki, has been the most widely used since 1999, although a new classification was proposed by Aydin and Harvey-Woodworth in 2014, which divides halitosis into five types: type 1 (oral), type 2 (airway), type 3 (gastroesophageal), type 4 (blood-borne), and type 5 (subjective). All of these features appear to be type 0 (physiological) in healthy people [5]. Seemann et al. recommended the use of simplified Miyazaki classification as an outcome of the international consensus workshop in 2014. It is advised to use following terms in diagnosis of halitosis in general dental practise: intra-oral halitosis (when the source of symptoms is inside the oral cavity), extra-oral halitosis (when the source is outside the oral cavity), temporary halitosis (when it is caused by dietary factors), pseudo-halitosis (when obvious malodor is not perceived by others but the patient stubbornly complains of its existence) and halitophobia (when the patient persists in believing to suffer from halitosis) [6].

The sources of halitosis are various; however, in most patients (>80%), it results from the activity of microorganisms in the oral cavity [2,3]. The bacteria most likely responsible for the production of hydrogen sulfide are those of the genera *Bacteroides*, *Eubacterium*, *Fusobacterium*, *Peptostreptococcus*, *Porphyromonas*, *Selenomonas*, and *Veillonella*, as well as *Tannerella forsythia* [7]. The most active producers of hydrogen sulfide are Gram-negative anaerobes such as *Porphyromonas gingivalis*, *Treponema denticola*, and *Tannerella forsythia*, i.e., red complex, important in the pathogenesis of periodontal diseases [7]. Another important volatile compound is methyl mercaptan, which is a metabolic product—mainly of *Porphyromonas* spp. and *Treponema denticola* [7]. Moreover, the number of bacteria such as *Prevotella*, *Alloprevotella*, and *Megasphaera* is significantly higher in patients with halitosis [8]. The types of *Prevotella* and *Alloprevotella* may also be associated with the production of the amines important in the pathogenesis of halitosis—putrescine and cadaverine [8].

Two large groups can be distinguished among the most common causes of halitosis: chronic gingivitis induced by biofilm and periodontitis; caries and its consequences in the form of pulp diseases; bacterial and fungal stomatitis; and inflammatory changes in the mucosa and bones; neoplastic and developmental changes in the mucosa and reduced secretion of saliva [2,3,4]. The second group consists of patients with extra-oral halitosis, because of laryngological causes, acute and chronic inflammation, tonsil stones, tonsil crypts, acute and chronic sinusitis, nasal polyps, nasal septum curvature, foreign bodies in the nasal passage, diseases of the salivary glands, and neoplastic lesions of the nasopharynx, sinuses, upper respiratory tract, and gastrointestinal tract [2,3,4]. Gastroenterological diseases can also cause extra-oral halitosis, including gastroesophageal reflux disease, gastric cancer, esophageal diverticulum, and *Helicobacter pylori* infection, as well as metabolic diseases, such as diabetes, renal and hepatic failure, trimethylaminuria, hypermethioninemia, and cystinosis [2,3,4]. The medications taken by the patient are also important; for example, antihistamines, phenothiazine derivatives, metronidazole, and bisphosphonates influence the formation of unpleasant odor from the mouth [4].

The diagnosis of halitosis is based on a carefully collected medical history, as well as quantitative and qualitative assessment. The organoleptic method, although it is a subjective method, is considered the gold standard, and consists of the direct assessment of the smell of the exhaled air 10 cm from the patient’s mouth, or the indirect assessment of the smell of saliva after wetting the wrist with it and waiting 10 s for it to dry [3,4,9]. Objective methods include gas chromatography, the BANA test, ammonia monitoring, the ninhydrin method, and the cysteine challenge test [3,4,9], which induces the production of H_2_S by bacteria present in the oral cavity. There are also portable devices useful in everyday work, enabling chairside objective assessment of parameters in patients a Halimeter with a volatile sulfur compounds sensor, or an Oral Chroma, which is based on gas chromatography [2,9].

## 2. Halitosis in Dental Prosthetics

In dental prosthetics, the problem of halitosis occurs both in patients using removable prosthetic restorations and in those who have fixed dentures cemented in the oral cavity. In both cases, new opportunities appear for the development of microorganisms, including those related to intra-oral halitosis. The review does not discuss dental implants.

### 2.1. Removable Dentures

Bacteria are able to form a biofilm on the surface of poly(methyl methacrylate) (PMMA) [10]—the main material used in the production of removable dentures. The ability to colonize and create biofilm on the surfaces of prostheses allows bacteria to use this additional niche for growth in the presence of a specific protective environment, which makes it difficult to remove them; hence, in cases of poor oral hygiene, prostheses can serve as a reservoir and cause persistent halitosis [10].

The incidence of volatile sulfur compounds, such as hydrogen sulfide, is higher among people who wear removable dentures compared to people who do not wear dentures, with hydrogen sulfide associated with-coated tongue and wearing-dentures overnight, while dimethyl sulfide is associated with systemic diseases and medications [11]. Patients with dentures (both removable and fixed) in the oral cavity also show higher levels of β-galactosidase (a parameter related to the level of volatile sulfur compounds in the oral cavity), and achieve higher results in the organoleptic and quantitative assessment of halitosis, with higher parameters observed in patients with removable dentures [12]. Many patients with removable dentures also complain of reduced salivation; this may be related to age, comorbidities, or medications. Reduced saliva flow, which causes the oral mucosa to dry out, is also one of the risk factors for mouth odor [13]. As saliva secretion decreases, its viscosity increases, and higher salivary viscosity may also be a potential risk factor for halitosis [13]. Dryness and burning of the mouth and failure to comply with the recommended nighttime break in using dentures are significantly correlated with the levels of volatile sulfur compounds in the oral cavity [14]. The unpleasant odor from the mouth in people wearing complete dentures can be reduced not only by maintaining good oral and denture hygiene, but also by following the recommended nighttime break in their use [15].

The effect of relining a removable denture on halitosis was also investigated. There was a significant reduction in the severity of halitosis immediately after relining compared to the state before relining, caused by the hygienization carried out before the procedure and the loss of the base material due to the preparation of the relined prosthesis for this procedure, simultaneously removing the bacteria associated with halitosis [16]. Research results suggest that prostheses that have been used by the patient for a long time are difficult to reline due to their potentially deep colonization by microorganisms and the methyl mercaptan secreted by them [17]. This compound causes the separation of relining material from the base material of the prosthesis, even after applying the primers to their surface. Methyl mercaptan is a chemical compound used to stop the elongation of the molecular chain in industrial polymerization processes [17].

### 2.2. Fixed dentures

Fixed dentures, despite the fact that they are a significantly different in construction comparing to removable dentures may also make it difficult to maintain proper oral hygiene—not only through the retention of food debris and the deposition of dental plaque [15], but also due to the inaccessibility to daily hygiene procedures and impaired ability of their self-cleaning by saliva [18]. The values obtained from a study with the Oral Chroma device showed the presence of halitosis in 65.9% of patients with dental crowns, and only in 32.69% without fixed dentures in the oral cavity [19]. The influence of the position of the crown edge, the marginal integrity, and the contour of the crown on the presence of halitosis was investigated. The parameters found to be defective included the subgingival margin, over-contouring, and crowns with leaking margin; the levels of volatile sulfur compounds were significantly higher in these patients compared to patients with non-defective crown parameters [19]. Thus, improperly made crowns significantly worsen the hygienic conditions and the microbiota of the oral cavity, resulting in a high frequency of halitosis [19]. However, it was found that hygienic procedures do not cause significant changes in the oral cavity until the etiological factor—for example, a low-quality fixed denture—is removed [18]. The reason for the association of the subgingival edge of the crown with subsequent gingivitis and halitosis may be the difficulty in removing all excess cement after dentures cementation on the abutment teeth—especially when using cements with adhesive properties [20]. Prosthetic bridges can also contribute to an unpleasant odor from the mouth. Therefore, tight structures with spans designed in such a way that it is possible to perform hygienic procedures and eliminate the risk of halitosis are necessary [18]. It is not only defective restorations—especially those that are poorly fitted—that may significantly contribute to the development of halitosis; additionally, in the case of their loosening, the problem may be exacerbated, because they allow the accumulation and proliferation of bacteria in the internal gaps between the abutment tooth and the restoration, creating a reservoir of anaerobic bacteria [15]. The material from which the fixed denture is made also matters; the highest values of volatile sulfur compounds were recorded in the case of restorations made of full metal, while the lowest values were recorded in the case of restorations made with CAD/CAM technology [20]. All ceramic restorations with supragingival margins showed the lowest degree of halitosis, followed by supragingival metal–ceramic restorations, while subgingival full-metal restorations showed the highest degree of halitosis [20].

## 3. Treatment of Halitosis

As previously mentioned, prosthetic restorations used by patients can affect the efficiency of hygienization due to the increased retention of food debris and plaque, as well as the difficulty of accessing the niches they create. It has been also noted that the increase in halitosis in denture wearers could be related to various factors, such as bacterial plaque on the tongue [12,14,21]. Therefore, the treatment of halitosis in prosthetic patients should not be limited only to the prosthesis itself if diagnosis confirms yet another source of this ailment. It has been observed that patients using both fixed and removable dentures show insufficient levels of oral hygiene, so it is necessary to educate them in the field of oral hygiene in general [22], and not merely in maintaining proper hygiene of dentures. Therefore, despite the fact that the topic of this article is halitosis in dental prosthetics, the authors found it justified to mention treatment methods used in patients who do not use prostheses, as they may also be beneficial in patients with prostheses.

### 3.1. Conventional Treatment

Conventional treatment of intra-oral halitosis is to reduce microbes via chemical treatment, with mouthwashes, and/or mechanically for example by a tongue scraper or a tongue brush [23]. Chemical ingredients used to control oral malodor include chlorhexidine, triclosan, cetylpyridine chloride, zinc ions, and chlorine oxide, as well as essential oils [24]. However, preparations containing chlorhexidine or combinations of chlorhexidine, cetylpyridine chloride, and zinc are the most effective [9,24]. Chemicals contained in mouthwashes, while providing effective and immediate relief, are only a temporary solution [23]. Mechanical methods have some disadvantages—they can cause a feeling of discomfort and induction of a gag reflex when performing hygienic procedures; there is also no standardized technique for performing them [23]. There is no conclusive scientific evidence for the effectiveness of conventional methods, and even when reports indicate a positive effect, the effect after their use is not permanent [23,25]. For the same reason, it is also impossible to state which of the above-mentioned methods is more effective in controlling halitosis [25]. There are no acceptable guidelines or protocols for the treatment of the oral odor and that may result from the variable etiology of this ailment [9,25,26].

### 3.2. Treatment in Prosthetic Patients

The effectiveness of agents commonly used for cleaning and improving the retention of dentures in reducing halitosis was tested. A commercial cleaning agent based on sodium perborate in the form of tablets, when the prosthesis was immersed in it once a day for 5 min, had no effect on the microorganisms of the prosthesis biofilm, and did not reduce the levels of volatile sulfur compounds [27]. Some denture adhesives containing sodium–magnesium–zinc and calcium–zinc copolymers such as PVM/MA, or lactoperoxidase, lysozyme, and lactoferrin, can inhibit microorganisms associated with halitosis [28]. 

Currently, the search for new strategies in the treatment of halitosis is underway; for example, one idea is to use photodynamic therapy, the advantages of which include high toxicity to bacteria in the absence of resistance, the absence of side effects, and the preservation of the physiological microbiota of the oral cavity [29]. This technique is performed with the use of light of a given spectrum and a photosensitizer appropriate for it. In a study using methylene blue and a 660 nm diode laser with six points on the tongue and six points on the prosthesis, it was found that the effect of photodynamic therapy on hydrogen sulfide levels was significant up to 30 days of observation; there was also a significant reduction in *Porphyromonas gingivalis*, but this did not last as long [29]. In another study, a red LED diode was used, thanks to which the number of irradiated points was reduced to four, and the PDT was immediately effective in reducing the emission of hydrogen sulfide from the oral cavity to a level consistent with the absence of halitosis, but the effect did not last after 7, 14, and 30 days of observation, nor did it cause a reduction in the numbers of *Porphyromonas gingivalis*, *Treponema denticola*, and *Tannerella forsythia* [30]. The effectiveness of photodynamic therapy using a blue LED in combination with *Bixa orellana* extract has also been tested, and appears to be effective in reducing halitosis immediately; however, similarly to the above-mentioned studies, this effect was not maintained in the seven-day control [31]. 

Another new approach to reducing the occurrence of halitosis is the attempt to modify poly(methyl methacrylate) (PMMA, which is used in the manufacture of removable dentures) with silver and graphene nanoparticles. Silver nanoparticles have a strong antimicrobial effect [32] and, in combination with graphene—a material with additionally improved mechanical properties and reduced water sorption—this effect is obtained at just 1% of their content [32,33]. Due to the addition of silver nanoparticles, the adhesion of *Candida albicans* decreased and the growth of bacteria on the PMMA surface was inhibited [34]. When it comes to the biocompatibility of PMMA after such modification, there are no statistically significant differences between PMMA disks with and without silver nanoparticles, but the synthesis method may have an impact [34]. In another study, nanographene oxide was used as an additive to PMMA in order to induce antimicrobial adhesive effects by increasing the hydrophilicity of the material; PMMA showed a better anti-adhesive effect against microorganisms for up to 28 days after such modification [35]. 

Additionally, attempts have been made to combine modified PMMA with photodynamic therapy. Light-activated PMMA nanofibers doped with a meso-tetraphenylporphyrin photosensitizer were created, the irradiation of which led to the migration and release of silver nanoparticles to the surface [34]. Photodynamic therapy using a photosensitizer applied to prostheses outside the mouth may be a safer, easier, and more effective procedure to inhibit the growth of oral bacteria in people wearing prostheses, without the risk of pain, mucosal lesions, or other side effects [32]. 

### 3.3. Other Supplementary Methods

Due to the fact that the main causative factor of intra-oral halitosis is dysbiosis in the composition of the commensal microbiota of the oral cavity, another method of treating halitosis is the use of probiotics [36,37]. *Streptococcus salivarius* is a non-pathogenic and dominant species that is an early colonizer of the oral cavity, and is one of the most important and most often isolated bacteria from people without halitosis [36,37,38,39,40,41]. Clinical studies are promising and show that the antimicrobial mouthwash containing *Streptococcus salivarius* K12 significantly reduced the number of volatile-sulfur-compound-producing bacteria [36]. In another study, *Streptococcus salivarius* K12 lozenges were used, and measurement of volatile sulfur compounds after their use also showed a significant improvement in most subjects.

## 4. Conclusions

The problem of the occurrence of ailments such as halitosis has accompanied patients for many years; this is no different in dental prosthetics. It should be noted that dentures—both fixed and removable—are a foreign body placed in the patient’s oral cavity which, in case of insufficient hygiene, may constitute a reservoir of microorganisms, causing this unpleasant ailment. The basis for the elimination of halitosis is proper diagnostics resulting in accurate diagnosis of the source of the problem and effective treatment, which in the case of intra-oral halitosis caused by insufficient hygiene should be comfortable enough for the patients that they decide to include it in their daily routine. Thanks to the latest research, the treatment does not have to be limited only to improving oral hygiene; it may also be supported by using new strategies, such as the modified denture material PMMA, photodynamic therapy, and/or probiotics.

## Data Availability

Data supporting our results are available for request from the corresponding author.

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
