# Peer review of "The Problem of Halitosis in Prosthetic Dentistry, and New Approaches to Its Treatment: A Literature Review"

_jcm, 2021, doi:10.3390/jcm10235560_

Round 1

Reviewer 1 Report

line 37 -39 the sentence is difficult to understand due to the grammar

Overall the text is sound written and well structured.

The given references are up to date and give an overview on the actual clinical situation as well.

There are some spelling mistakes in the text, such as:

spelling mistake line 47 "hesuffers"

line 64 capitalized "A"

line 170 "e.g."

Please use consequent : abbreviations for pathogens such as Peptostreptococcus etc or as full word throughout the text

Author Response

Dear Reviewer,

        I would like to thank You for this  exact and conscientious opinion. It has many valuable points for us to take into account and make changes.  These comments are for very important so we can make a lot of improvements in our study. Thank You for concrete details helping us to make this study more valuable.

line 37 -39 the sentence is difficult to understand due to the grammar

Changed the sentence from:

Halitosis (lat. fetor ex ore, fetor oris)  a condition in which an unpleasant odor comes out of the mouth, which is caused by e.g. volatile compounds of sulfur such as hydrogen sulfide, methyl mercaptan and dimethyl sulfide, as well as other vola-tile compounds such as indole, skatole, acetic acid, propionic acid and amines (ca-daverine and putrescine).

To:

Halitosis (lat. fetor ex ore, fetor oris)  is a condition in which an unpleasant odor comes out of the mouth. It is caused by volatile compounds of sulfur such as hydrogen sulfide, methyl mercaptan, dimethyl sulfide, indoles, skatoles, acetic acid, propionic acid and amines.

Overall the text is sound written and well structured.

The given references are up to date and give an overview on the actual clinical situation as well.

There are some spelling mistakes in the text, such as:

spelling mistake line 47 "hesuffers" – changed to „he suffers”

line 64 capitalized "A" – changed to „and” (unnecessary capital letter)

line 170 "e.g." – removed „e.g”, was there by mistake

Please use consequent : abbreviations for pathogens such as Peptostreptococcus etc or as full word throughout the text

Changed the names of pathogenes for full word throughout the text.

Reviewer 2 Report

Abstract

  • The aim of the present study should be clearly described in this section.
  • Halitosis is also caused by non sulfured compounds. This must be included in the Abstract.
  • The terms “unconventional solutions/methods” are not clear. Did the authors mean “new strategies?”
  • The sentences here may be re arranged. Authors are encourage to firstly inform the data from observational studies and, afterwards, the data from interventional studies. In its current format, it is very difficult to understand.
  • Regarding the information of “new treatments”, it is not clearly described the relationship with prosthetic dentistry. The absence of a clear aim for the present study might explain it. Did the author want to review the literature about the treatment of halitosis in general or only in those with dental prosthesis?

Introduction

    • The following sentence is not clear: "This is the most widely used Mcchin classification by Miyazaki and the 1999 century.” What is “the 1999 century?”
    • Please clarify the classification of Aydin-Woworth. Both last sentence of this paragraph are very difficult to read.
    • Both sentences are repetitive in this section: "have the ability of intra- or extraoral” and "The etiology of pathological halitosis may be intra- or extraoral.”
    • Italicized all bacteria species.
    • Replace to "Porphyromonas spp. and Treponema denticola [6].”
    • The term “gingivitis induced by biofilm” and “periodontitis" are recommended by the current Classification of the periodontal diseases.
    • Prefer the term “calculus" instead of “tartar.”
    • Cysteine challenge test is used to “induce" halitosis and its treatment. Please align it.

Halitosis and dental prosthetics

  • In this section, the cited studies are cross-sectional ones, which does not allow temporality. Therefore, the following sentence is inappropriate: “Therefore, prosthetic crowns intensify the development of halitosis."

Treatment of halitosis

  • In order to allow a proper alignment with the rest of the manuscript, this section must be restricted to the halitosis of those using dental prosthesis. Oral procedures that do not include oral prosthesis must be synthesized and/or excluded. If that is not the case, a proper aim must be disclosed.
  • This section is formed by a very long paragraph. This is very difficult to read. Present each idea in one paragraph.

Conclusions

- No aspects related to the treatment of halitosis in those using dental prosthesis is presented in this section.

Author Response

Dear Reviewer,

         I would like to thank You for this precisely done review.  All the tips are very valuable for us. We explained indicated issues more precisely and changed references in the introductuon. Thank You for concrete details helping us to make this study more valuable. Making changes in this manuscript according to Your comments was an opportunity for us of gaining knowledge and great experience.

Abstract

The aim of the present study should be clearly described in this section.

Added sentence with the aim of this work:

Line 17–18 – „The aim of this work a review of recent information about halitosis among patients using prosthetic dentures.”

Halitosis is also caused by non sulfured compounds. This must be included in the Abstract.

Line 18 - Added „non sulfured”

The terms “unconventional solutions/methods” are not clear. Did the authors mean “new strategies?”

Line 23 – changed „unconventional solutions” to „new strategies”

The sentences here may be re arranged. Authors are encourage to firstly inform the data from observational studies and, afterwards, the data from interventional studies. In its current format, it is very difficult to understand.

Sentences from line 26-30: „The problem of the occurrence of halitosis has accompanied patients for many years. It should be realized that dentures, both fixed and removable, are a foreign body placed in the patient's oral cavity, which, in case of insufficient hygiene, may constitute a reservoir of microorganisms causing this unpleasant condition. „ are now rearranged. The first sentence from this section is in line 19-20 and the rest of it is in line 22-24, before the data from the interventional studies.   

Regarding the information of “new treatments”, it is not clearly described the relationship with prosthetic dentistry. The absence of a clear aim for the present study might explain it. Did the author want to review the literature about the treatment of halitosis in general or only in those with dental prosthesis?

Added clear aim of this study at the beggining of the introduction;

Line 17–18 -  added sentence:  „The aim of this work a review of recent information about halitosis among patients using prosthetic dentures.”

Introduction

The following sentence is not clear: "This is the most widely used Mcchin classification by Miyazaki and the 1999 century.” What is “the 1999 century?”.

Changed the sentence from:

„This is the most widely used Mcchin classification by Miyazaki and the 1999 century.”

To:

„This is the most widely used classification by Miyazaki from 1999.” (line 48)

Please clarify the classification of Aydin-Woworth. Both last sentence of this paragraph are very difficult to read.

Line 49 - Changed the name of classification (spelling mistake), for appropriate:

„A new classification by Aydin and Harvey - Woodworth in 2014………”

Line 49-54 – changed sentences according to the reviewer’s suggestions for:

„This is the most widely used classification by Miyazaki from 1999, although there is a new classification by Aydin and Harvey-Woodworth from 2014, and it divides halitosis into five types: type 1 (oral), type 2 (from the airway), type 3 (gastroesophageal),  type 4 (blood-borne) and type 5 (subjective). All these features appear to be type 0 (physiological) also in healthy people [5].”

Line 54 - we’ve found a mistake „potassium” was changed to „physiological”, which is correct here.

Both sentences are repetitive in this section: "have the ability of intra- or extraoral” and "The etiology of pathological halitosis may be intra- or extraoral.”

Line 54 – Changed the sentence: "The etiology of pathological halitosis may be intra- or extraoral.” to „The sources of halitosis are various.”

Italicized all bacteria species.

Changed the names of bacteria species for italics.

Replace to "Porphyromonas spp. and Treponema denticola [6].”

Line 62-63 – Changed into  "…Porphyromonas spp. and Treponema denticola [6].”

The term “gingivitis induced by biofilm” and “periodontitis" are recommended by the current Classification of the periodontal diseases.

Line 69-70 – according current classification, changed for:
„…..gingivitis induced by biofilm and periodontitis….."

Prefer the term “calculus" instead of “tartar.”

Line 73 – changed „tartar” to „calculus”

Cysteine challenge test is used to “induce" halitosis and its treatment. Please align it.

Line 93 –  Added:  „… which induces the production of H2S by bacteria present in the oral cavity..”

Halitosis and dental prosthetics

In this section, the cited studies are cross-sectional ones, which does not allow temporality. Therefore, the following sentence is inappropriate: “Therefore, prosthetic crowns intensify the development of halitosis."

Line 133-134 – removed the sentence: “Therefore, prosthetic crowns intensify the development of halitosis."

Treatment of halitosis

In order to allow a proper alignment with the rest of the manuscript, this section must be restricted to the halitosis of those using dental prosthesis. Oral procedures that do not include oral prosthesis must be synthesized and/or excluded. If that is not the case, a proper aim must be disclosed.

This section is formed by a very long paragraph. This is very difficult to read. Present each idea in one paragraph.

 Added a new paragraph in lines 173-185 to justify including other methods of treatment:

 „As previously mentioned, prosthetic restorations used by patients -can affect the efficiency of hygienization due to the increased retention of food debris and plaque, as well as difficult access to the niches created by them. It has been also noted that the increase of halitosis in denture wearers could be related to various factors, such as bacterial plaque on the tongue (11, 13, 20). Therefore, the treatment of halitosis in prosthetic patients should not be limited only to the prosthesis itself, if diagnosis confirm yet another source of this ailment. It has been observed that patients using both fixed and removable dentures show insufficient levels of oral hygiene, so it is necessary to educate them in the field of oral hygiene in general (21), not only about maintaining proper hygiene of dentures. Therefore, despite the fact that the topic of this article is halitosis in dental prosthetics, the authors found it justified to mention treatment methods used in patients who do not use prostheses, as they may also be beneficial in patients with prostheses.”

We added another 2 items in references (numbers 20 and 21 in lines 367-372).

According to Reviewer’s suggestion, each idea is now presented in separate paragraph.

Added „ speciffically in prosthetic” in line 242 and „which is used in removable dentures manufacturing” in lines 243-244 to make it more understandable.

Conclusions

No aspects related to the treatment of halitosis in those using dental prosthesis is presented in this section.

Changed the conclusion, added and changed last sentences:

„The basis for its elimination is proper diagnostics resulting in accurate diagnosis of the source of the problem and effective treatment, which in the case of oral halitosis caused by insufficient hygiene should be comfortable enough for the patient that he decides to include it in the daily routine. Thanks to the latest research, the treatment does not have to be limited only to improving oral hygiene. It may be supported by using new strategies like modified denture material PMMA, photodynamic therapy and probiotics.”

Extra corrections:

We’ve found mistakes in translation here: „city halitosis” in the line 42 and „true halitosis” in the line 43  were changed to „genuine halitosis”.

Line 36-39  -  changed the beginning of introduction for: 

"Nowadays, in the age of broadly understood social media and the aesthetic standards they create, and the great need to be socially accepted by meeting them, many people also struggle with exclusion due to halitosis, i.e. the odor from the mouth. The prevalence of this ailment in the general population is as high as 31.8% [1]."
